# Results of the First Folate Receptor Alpha Testing Trial by the German Quality Assurance Initiative in Pathology (QuIP^®^)

**DOI:** 10.3390/cancers17223703

**Published:** 2025-11-19

**Authors:** Alexander Scheiter, Sven Mattern, Verena Gassenmaier, Hans-Ulrich Schildhaus, Matthias Christgen, Hans Kreipe, Hermann Herbst, Bettina Lambert, Guido Sauter, Maximilian Lennartz, Korinna Jöhrens, Florian Sperling, Afschin Soleiman, Ramona Erber, Stephan Singer, Annette Staebler, Kirsten Utpatel

**Affiliations:** 1Institute of Pathology, University of Regensburg, 93053 Regensburg, Germanykirsten.utpatel@klinik.uni-regensburg.de (K.U.); 2Department of Pathology and Neuropathology, University of Tübingen, 72076 Tübingen, Germanyannette.staebler@med.uni-tuebingen.de (A.S.); 3Institute of Pathology Nordhessen, 34119 Kassel, Germany; 4Institute of Pathology, Hannover Medical School, 30625 Hannover, Germany; 5Department of Pathology, Vivantes Klinikum Neukölln, 12351 Berlin, Germany; 6Institute of Pathology, University Medical Center Hamburg-Eppendorf, 20246 Hamburg, Germany; 7Quality Assurance Initiative Pathology (QuIP GmbH), 10117 Berlin, Germanysperling@quip.eu (F.S.); 8INNPATH, Institute of Pathology, Tirol Kliniken Innsbruck, 6020 Innsbruck, Austria

**Keywords:** folate receptor alpha (FRα), immunohistochemistry (IHC), antibody-drug conjugates (ADC), proficiency testing, biomarker validation

## Abstract

Reliable testing of folate receptor alpha is essential to identify patients who may benefit from a targeted treatment for ovarian cancer. In Europe, laboratories currently may use different antibodies and staining systems, but it is not known whether these approaches provide comparable results. In this study, we conducted a large quality assessment to examine how well different laboratories and methods can detect folate receptor alpha in ovarian cancer samples. We found that the currently approved test (companion diagnostics by Roche VENTANA) showed the most consistent performance, while several widely used alternative antibodies often produced weak staining or false positive results. Our work highlights the importance of conducting open proficiency trials to evaluate alternative biomarker testing approaches.

## 1. Introduction

Folate receptor alpha (FRα) is a membrane-bound glycoprotein anchored by glycosyl-phosphatidylinositol (GPI) and encoded by the *FOLR1* gene [1]. FR plays a pivotal role in several processes fundamental to tumorigenesis, such as DNA synthesis, cell proliferation, DNA repair, and intracellular signaling pathways [2]. Upon binding with folate, FRα triggers a cascade of intracellular signaling through phosphorylation, which activates pathways like ERK and STAT3, essential for regulating cell growth mechanisms. Additionally, FRα aids in tumor invasion and metastasis by reducing the expression of adhesion molecules, including E-cadherin [3]. FRα is found in a significant proportion of various cancers, including ovarian cancer, non-small cell lung cancers, endometrial cancers and triple-negative breast cancers. In comparison, FR expression in non-cancerous tissues is limited; for instance, in non-neoplastic ovaries, expression was either absent or only weakly detected [4,5]. However, variable, moderate-to-strong mebrane expression is observed in fallopian tube epithelium, which is recommended as a positive control by the Ventana FOLR1 RxDx Assay. This cancer-specific and organ-specific expression of FR allows for targeted treatments—such as antibody-drug conjugates—that can selectively target cancer cells while sparing healthy tissue.

The U.S. Food and Drug Administration (FDA) approved mirvetuximab soravtansine-gynx, an antibody-drug conjugate, on 22 March 2024, for treating adult patients with FRα-positive, platinum-resistant epithelial ovarian, fallopian tube, or primary peritoneal cancer [6]. These patients had to have received one to three prior systemic therapies. This approval was accompanied by the VENTANA FOLR1 (FOLR-2.1) RxDx Assay, an immunohistochemical test used to identify FRα-positive tumors through moderate-to-strong membrane staining in at least 75% of tumor cells. This test serves as a companion diagnostic to select appropriate patients for treatment with mirvetuximab soravtansine-gynx. The approval was based on findings from the MIRASOL study, which involved patients with high FRα-expressing, platinum-resistant cancers who had undergone one to three previous systemic therapies. Median overall survival was 16.5 months (95% CI: 14.5, 24.6) in the mirvetuximab soravtansine-gynx arm and 12.7 months (95% CI: 10.9, 14.4) in the chemotherapy arm (Hazard Ratio [HR] 0.67 [95% CI: 0.50, 0.88] *p*-value 0.0046). Median progression-free survival was 5.6 months (95% CI: 4.3, 5.9) and 4.0 months (95% CI: 2.9, 4.5) (HR 0.65 [95% CI: 0.52, 0.81] *p*-value < 0.0001) for the respective arms. Overall response rate was 42% (95% CI: 36, 49) and 16% (95% CI: 12, 22) (*p*-value < 0.0001) [7].

On 14 November 2024, the European Medicines Agency (EMA) granted marketing authorisation for mirvetuximab soravtansine (Elahere), intended for the treatment of adults with FRα-positive epithelial ovarian, Fallopian tube and primary peritoneal cancer [8].

Unlike in the United States, Europe continues to maintain flexibility in the choice of antibodies and detection systems. In this context, a proficiency test by the Quality Assurance Initiative for Pathology (QuIP^®^) was launched to equip the German-speaking pathology community with the tools needed for a reliable and reproducible immunohistochemical biomarker test. In this study, we report the results and experiences of the first folate receptor alpha proficiency test.

## 2. Materials and Methods

QuIP nominated two lead panel institutes to conduct the first folate receptor alpha (FRα) proficiency test, specifically the Institute of Pathology at the University Hospital Tübingen and the University of Regensburg. Additional panel institutes were named in Hamburg, Hanover, Berlin, Kassel, and Innsbruck. The lead panel institutes were responsible for case selection and execution of the proficiency test. Between July and September 2024, ovarian carcinomas from the archives of the two lead institutes were screened for FRα expression.

For the internal proficiency test, 25 ovarian carcinoma cases that had already been validated by the lead institutes were selected (Regensburg: *n* = 12, Tübingen: *n* = 13). Only those tumors that showed no major discrepancies between the two reference centers when tested with the VENTANA FOLR1 (FOLR1-2.1) RxDx Assay were included. Cases were considered concordant when consistently classified as FRα-positive or FRα-negative; minor deviations in the percentage of positive tumor cells were tolerated, provided that they did not occur in the immediate vicinity of the clinically relevant cut-off of 75%. These 25 validated cases were subsequently distributed to the five additional panel institutions. The participating centers were granted methodological freedom and were not restricted to the Ventana RxDx assay; instead, they were allowed to apply their own antibodies and staining platforms. The Ventana RxDx assay nevertheless served as the study’s gold standard, with positive cases defined according to its results. Case selection was therefore not limited exclusively to strictly concordant results in order to avoid bias towards very homogeneously expressing tumors. Instead, the overall reproducibility of the Ventana RxDx assay, as confirmed by the reference centers, served as the guiding criterion.

Panel Institute 1 demonstrated that 23 out of the 25 cases yielded concordant results when tested with a laboratory-developed test (LDT) based on the BN3.2 antibody clone (Novocastra). From these concordant results, eight cases were selected that were consistently evaluated as concordant by both the lead institutes and at least four of the panel institutes. In addition, two cases were included in which concordance was achieved exclusively with the Ventana FOLR1 RxDx Assay but not with alternative assays. An overview of the applied antibodies and staining procedures by the lead panel and panel institutes is provided in Table 1. For the Roche VENTANA FOLR1 [FOLR1-2.1] RxDx Assay, incubation time strictly followed the manufacturer’s instructions to ensure compliance with the companion diagnostics protocol. As this antibody is supplied ready-to-use, no dilution adjustments were required. For all other antibodies, incubation times were individually determined by each participating institution. While general recommendations are provided in the respective antibody data sheets, these reagents are not IVD-approved, and therefore, laboratories performed gradient testing and protocol optimization according to their staining platforms and detection systems. This optimization typically involved fine-tuning antibody dilution, incubation time, and detection chemistry using appropriate positive control tissues. Consequently, some variability in incubation times and dilutions existed across sites reflecting real-world conditions.

The open proficiency test was organized in two splits. Split 1 was conducted from 9 December 2024 to 20 December 2024, while Split 2 was carried out from 17 March 2025 to 4 April 2025. Split 1 was based on the ten cases selected from the original pool of 25 validated cases, according to the predefined concordance criteria described above. A flowchart depicting the overall design of the internal and open proficiency trials can be found in Figure 1.

Split 2 was largely identical to Split 1 but required the introduction of two replacement cases due to tissue exhaustion in some of the original samples. These replacement cases were also drawn from the initial 25 cases validated by the lead panel institutes and were specifically chosen to mirror the same FRα expression patterns as the depleted samples. In total, 12 unique cases were therefore used across the two splits. To maintain consistency and avoid potential bias, the sequence of cases in Split 2 was adjusted, ensuring that all participants received material in a nonidentical order. For clarity, all tabular and graphical overviews presented in this study follow the case order from Split 2, which is applied consistently throughout the manuscript.

FRα expression in epithelial ovarian cancer (EOC) tissues was evaluated using both staining intensity and percentage of stained viable tumor cells. EOC was considered positive for FRα expression when 75% or more of the viable tumor cells exhibited moderate or strong membrane staining intensity. Membrane staining may be apical or circumferential (partial or complete). Also, a dot-like staining pattern with dark brown secretions filling the gland lumens is included in scoring. This scoring method was adopted based on data from prior mirvetuximab soravtansine clinical trials [7,9].

For each correctly evaluated case, two points were awarded. With 10 cases, this resulted in a total score of 20 points. In the event of technical issues (marked as “Technically not assessable”), the lead panel institute reviewed the average score retrospectively and decided on point allocation based on the cause of the issue. To successfully complete the proficiency test, a minimum of 18 points was required. To assess interobserver variability, all submissions from participating laboratories were evaluated across the ten cases shown in Figure 2. Technically not assessable (TNA) results and values given in parentheses were excluded from the analysis.

For each case, the reference (target) classification (positive or negative) was defined by the lead panel institutes (cases 1, 2, 3, 4, 8, 9 = positive; cases 5, 6, 7, 10 = negative).

All individual case ratings from participants were combined into a pooled 2 × 2 contingency table comparing the reference classification with the corresponding participant rating (positive/negative). The observed agreement (*Pₒ*) was calculated as the proportion of correct ratings among all valid ratings.

The expected agreement by chance (Pₑ) was calculated from the marginal proportions of both raters (reference and participants) according to$$P_{e}=\sum_{k}p_{\left\{A,k\right\}}\cdot p_{\left\{B,k\right\}}$$
where $p_{\left\{A,k\right\}}$ and $p_{\left\{B,k\right\}}$ represent the marginal probabilities for category k ∈ {positive, negative}. Cohen’s kappa was then computed as$$\kappa =\frac{P_{o}-P_{e}}{1-P_{e}}$$

## 3. Results

### 3.1. Internal Proficiency Test by 2 Lead and 5 Panel Institutes

The VENTANA FOLR1 RxDx Assay (Roche Diagnostics), alongside its use by two lead institutes, was also utilized by an additional panel institute. In comparison, four other panel institutes employed alternative antibody clones: BN3.2 from Leica Biosystems (Newcastle upon Tyne, UK) and Novocastra (Newcastle upon Tyne, UK), a polyclonal rabbit antibody from Invitrogen (Carlsbad, CA, USA) and 26B3.F2 from Biocare (Pacheco, CA, USA).

When exclusively using the VENTANA FOLR1 RxDx Assay, a high concordance rate of 96% (24/25 cases) was achieved, with perfect sensitivity at 100% and strong specificity at 97.9%. Conversely, the use of all five antibody clones resulted in a lower concordance rate of 52% (13/25 cases). Notably, an identical result to the Ventana clone could also be demonstrated in 22 out of 25 cases using the BN3.2 antibody clone purchased from Novocastra. For the protocols of panel institutes 3 and 4, which applied the polyclonal antibody from Invitrogen and the BN3.2 clone purchased from Leica, a tendency towards reduced sensitivity and weaker staining intensity was observed, resulting in discordant negative classifications. In contrast, the protocol of panel institute 2 using the 26B3.F2 clone from BioCare Medical demonstrated strong staining intensity, which led to discordant positive classifications, i.e., false-positive cases, and consequently low specificity. The false-positive interpretation was caused by pronounced non-specific background staining, which obscured the distinction between true membrane staining and background signal, thereby complicating accurate evaluation. The results for the cases that were eventually used for the open proficiency trials are summarized in Figure 2.

It should be noted that cases 3b and 4b were replacement cases for Split 2 and were not used to determine success for the participants of the internal proficiency test. Instead, the remaining 10 cases were used for scoring. An overview of all cases assessed in the internal proficiency test is provided in Appendix A (these were, however, not further distributed for the open proficiency tests).

### 3.2. Open Proficiency Test Splits 1 and 2

Results of Splits 1 and 2 are analyzed together in the following section. A total of 70 registrations were received, with 22 registrations for split 1 and 48 for split 2. Institutes that did not successfully complete the internal proficiency test or split 1 were offered the opportunity to participate again in split 2, a possibility utilized by nine institutes, including two that had originally served as panel institutes in the internal proficiency test. Altogether, 63 distinct institutes participated in the open proficiency test (split 1 and/or split 2), in addition to five institutes that were involved exclusively in the internal proficiency test. Among the participants were 59 institutes from Germany, 7 from Austria, and 2 pathology departments from Switzerland. Of these, 28 were academic institutions, whereas 40 represented non-academic centers (including private pathology practices, municipal hospitals, and medical care centers/MVZ).

In the external proficiency testing scheme, each participating institution received a set of ten ovarian carcinoma cases prepared as 2 μm FFPE tissue sections. The participants were given 15 working days to perform immunohistochemical staining for FRα and to submit their evaluations. All submissions were compared with the reference values established by the two lead panel institutes, and an additional review process was performed to address potential technical failures or interpretative problems. This review was also used to guide the final scoring of results. After review of both splits, 41 of 68 institutions were classified as successful, corresponding to an overall success rate of 60%, which was similar to the success rate of the internal trial (57%). It should be noted that repeated submissions were possible in Split 2, allowing institutions that had either failed in the internal trial or had participated in Split 1 to resubmit their results in Split 2 (Table 2).

In several instances, points were awarded after review despite initial deviations from the reference values. For case 1, two submissions were affected by tumor heterogeneity and were confirmed as negative in the review despite adequate staining quality. For case 3, six institutions received points retrospectively; all of these had analyzed the replacement case 3b, which was more challenging to interpret than the original case and in some slides showed tissue artifacts such as cracks that complicated evaluation. Additional adjustments were made for six further submissions due to individual material heterogeneity in cases 4 and 6 or technical problems in cases 2, 5, 8, and 10, all of which had been documented by the participants and were judged acceptable. Taking these adjustments into account, 14 institutions achieved the maximum score of 20 points, and 23 additional laboratories were considered successful despite a deviation in one case. The remaining institutions did not reach the required threshold, as they deviated from the reference values in two or more cases, resulting in the final score values displayed in Appendix A.

When considering the methods applied, substantial differences in performance became apparent (Table 3). Institutions that used the VENTANA FOLR1 (FOLR1-2.1) RxDx Assay, achieved the highest concordance with the predefined reference values. In total, 30 of 36 participants (83%) using this assay reached a successful outcome. In contrast, the performance of the BN3.2 antibody clone was considerably weaker. Among the 23 institutions that used the Leica-supplied BN3.2 clone, only 22% were successful, and among eight institutions using the Novocastra version, 25% achieved a successful outcome. Three individual participants used BN3.2 from Cell Signaling, the EPR20277 antibody from Abcam, or a polyclonal anti-FRα antibody from Invitrogen, and none of these submissions were successful.

A more detailed case-by-case problem analysis (Table 3) revealed different dominant sources of error depending on the antibody used. For the Roche assay, interpretative difficulties were the main cause of failure. In 22 instances, misclassification could be traced back to interpretation, while only one case was due to insufficient staining and one to a combined staining and interpretative problem. By contrast, the BN3.2 clone was predominantly affected by staining-related problems. Here, 18 of 23 failed Leica antibody submissions and four of eight failed Novocastra antibody submissions showed insufficient staining intensity, particularly in the positive cases 1, 3, and 9. These cases consistently revealed a weaker staining reaction compared with the Roche reference assay, resulting in false-negative assessments.

Case-based analysis highlighted specific challenges across institutions (Figure 3). The majority of discordances clustered in cases 1, 3, 9, and 10, while the remaining cases were largely concordant. In cases 1, 3, and 9, false-negative results predominated, most often associated with BN3.2 protocols. Representative pictures of all cases that were correctly evaluated by participants using the VENTANA FOLR1 (FOLR1-2.1) RxDx Assay are shown in Appendix A. Here, also representative images of on-slide tubal epithelium-positive controls are displayed, which show strong (3+) apical membranous staining in epithelial cells, predominantly moderate (2+) circumferential staining, and no staining (0) in the stromal compartment.

To assess interobserver reliability, all submissions from participating laboratories were evaluated across the ten cases. Technically not assessable (TNA) results and values given in parentheses were excluded from the analysis.

The pooled analysis of all submissions against the reference classification showed an observed agreement of 83.4% and an expected agreement by chance of 49.7%, corresponding to Cohen’s κ = 0.67 (substantial agreement). The analysis of all submissions using the VENTANA FOLR1 (FOLR1-2.1) RxDx Assay showed an observed agreement of 88.5% and an expected agreement by chance of approximately 51%, corresponding to Cohen’s κ = 0.77 (substantial agreement). In contrast, the analysis of all submissions using the BN3.2 or EPR20277 antibodies showed an observed agreement of 77.9% and an expected agreement by chance of 48.4%, corresponding to Cohen’s κ = 0.57 (moderate agreement).

This finding was consistent across manufacturers and platforms and reflected the overall lower sensitivity of the other clones. The low sensitivity of clones other than VENTANA FOLR1 (FOLR1-2.1) is exemplarily shown in Figure 4 based on submissions regarding case 1. Another example regarding the ambiguous case 3a is shown in Appendix A. Moreover, we tested the observation of decreased sensitivity of the Leica BN3.2 clone as opposed to VENTANA FOLR1 (FOLR1-2.1) in a self-made tissue microarray of low- and high-grade serous carcinomas. Indeed, the same issue was encountered here, and also an increase in antibody concentration (i.e., lower dilution) did not remedy the issue, but instead resulted in a higher cytoplasmic background and increased difficulty in interpreting the actual membrane staining. Exemplary pictures highlighting this issue in an additional tissue microarray are displayed in Appendix A.

By contrast, case 10 frequently yielded false-positive interpretations. In this case, relatively strong cytoplasmic staining was present without a corresponding membranous reaction. Since the scoring guidelines required membranous positivity, such cases should have been categorized as negative, but cytoplasmic signals were erroneously interpreted as positive (Figure 5a). A further interpretative challenge was the recognition of dot-like expression patterns in narrow lumina. All tumor cells surrounding such luminal dot-like positivity should have been classified as positive, but several laboratories underestimated the proportion of positive tumor cells, which led to false-negative categorization (Figure 5b).

The review of all submissions (Figure 5c) showed that 61% of failures were due to staining quality, 18% to interpretation, and 21% to a combination of both. In line with the case-based analysis, staining issues were predominant among BN3.2 protocols, while interpretation errors were the leading cause of failure for the Roche assay.

The evaluation of the proficiency test also considered the use of the Roche antibody as a laboratory-developed test (LDT), i.e., deviations from the protocol of the VENTANA FOLR1 (FOLR1-2.1) RxDx Assay. Five participants applied the FOLR1 antibody outside of the approved Ventana platform, using alternative devices such as the Leica Bond-III or the Agilent Dako Omnis. Of these five, three were successful, corresponding to a success rate of 60%. In comparison, when the assay was applied as an IVD companion diagnostic on the Ventana BenchMark platform, the success rate was substantially higher at 87% (27 of 31 participants). Thus, while selected LDT protocols were able to achieve acceptable performance, the IVD assay consistently yielded more reliable results. A breakdown of the LDT protocols is provided in Appendix A.

Successful submissions were documented across a variety of automated platforms, including Leica Bond-III, Agilent Dako Omnis, and Zytomed IntelliPATH. Nevertheless, it was remarkable that none of the six institutions using the BN3.2 clone on a Ventana platform achieved a successful outcome (Appendix A).

To support the participants, a 60 min online seminar was held on October 10, 2024, before the internal proficiency test and before Split 1 and 2 of the external trial. This seminar addressed both staining procedures and interpretative challenges of FRα immunohistochemistry and included the joint review of five training cases. Among the 68 institutions that participated in the trials, 23 (34%) attended this seminar. The success rate among these institutions was 71% (17 of 23), whereas only 71% (15 of 21) of institutions without prior training were successful. This suggests that training contributed to improved recognition of interpretative pitfalls such as dot-like expression patterns or cytoplasmic staining artifacts and underlines the potential value of preparatory educational interventions.

## 4. Discussion

This prototype proficiency test represents the first systematic comparison of different antibodies for FRα immunohistochemical testing. The VENTANA FOLR1 (FOLR1-2.1) RxDx Assay consistently produced high concordance rates in both the internal and open proficiency tests, reinforcing its role as the gold standard for FRα detection. In contrast, alternative antibodies, particularly BN3.2 (Leica Biosystems or Novocastra) and EPR20277 (Abcam), exhibited significantly lower success rates, primarily due to weak staining intensity in critical cases. Given that most participants already used low antibody dilutions (1:50 or 1:25) and one participant, apparently aware of the lower sensitivity, even went down to 1:10 and incubated for one hour still being unsuccessful, further dilution or incubation time adjustments are unlikely to improve performance.

Our findings supplement and in part corroborate two recent studies evaluating FRα immunohistochemistry in ovarian cancer: Zannoni et al. [10] demonstrated in a multicentre analysis using the VENTANA FOLR1 (FOLR1-2.1) RxDx Assay that interobserver reproducibility of FRα scoring is generally high while the estimation of percentage of positive tumor cells, although borderline cases near the 75% cut-off remain a source of interpretative variability. This aligns with our observation that the Roche assay performed robustly across institutions, but that interpretative challenges persisted in specific situations such as dot-like luminal staining or cytoplasmic reactivity. Complementary evidence was provided by Deutschman et al. [11], who systematically compared six alternative FRα antibodies with the approved companion diagnostic and found that most non-Roche clones lacked sufficient specificity or membrane staining quality, while even the best-performing antibodies tended to overestimate FRα positivity relative to the VENTANA FOLR1 (FOLR1-2.1) RxDx Assay. These findings parallel the outcome of our proficiency test, in which non-IVD-labeled clones such as BN3.2 or EPR20277 showed markedly lower concordance and predominantly staining-related failures. Unexpectedly though, in this study the Leica BN3.2 overestimated the expression of FRα compared to the Ventana antibody. Given, that a major source of error when using VENTANA FOLR1 (FOLR1-2.1) RxDx Assay were interpretation issues, the value of adequate training should be noted, which in this study was ensured and evidenced by the offer of a seminar before the participation. Seminar attendance reduced mistakes due to misinterpretation of, for example, cytoplasmic and dot-like staining and was associated with higher pass rates.

In other studies examining the comparability of different biomarkers, such as PD-L1 in non-small cell lung cancer (NSCLC), or EGFR immunohistochemistry in squamous NSCLC, significantly higher concordance between antibody clones has been observed [12,13]. One possible explanation for the discrepancies in the present study could be that the Ventana antibody clone only became commercially available in Germany in December 2024. As a result, in-house comparative analyses with other antibody clones were not feasible at an earlier stage.

The discrepancies observed with alternative antibodies suggest that not all FRα-targeting assays are interchangeable, emphasizing the need for standardized biomarker assessment. This is particularly relevant for clinical settings where FRα expression determines eligibility for targeted therapies.

One major advantage in Europe is the freedom of methodology. In clinical studies, biomarkers were centrally tested, which made it easier to use the same antibody with the same detection system. However, in countries like Germany, where biomarker testing is decentralized and pathology labs typically use only one detection system, mandating a specific assay (including both the antibody and its corresponding detection system) could limit broad patient access. Finally, it is noteworthy that in some cases participants were successful using the FOLR1 antibody in an LDT setting with deviations such as the use of a different staining platform. Given that the Ventana antibody is several times more expensive than other available antibodies, its routine use is not economically feasible within the German reimbursement system. Therefore, affordable alternative clones are essential to ensure at least cost-covering implementation in routine diagnostics.

Altogether our findings highlight that it is crucial to evaluate reliable alternatives to the established assay—currently the Roche assay—through further external quality assessment (EQA) studies.

A key limitation of our study is the relatively small sample size for some antibody-detection system combinations. Future studies should investigate whether protocol adjustments, including antigen retrieval methods and antibody concentrations, could enhance the performance of alternative clones.

## 5. Conclusions

Our findings reinforce that FRα IHC testing should be performed using validated assays, such as the VENTANA FOLR1 RxDx Assay, to ensure reliable patient stratification for FRα-targeted therapies, while validated LDTs using the FOLR1 Roche antibody may also be acceptable. The variability observed with alternative clones underscores the importance of quality assurance measures, including proficiency tests, to maintain high diagnostic accuracy in clinical practice.

## Figures and Tables

**Figure 1 cancers-17-03703-f001:**
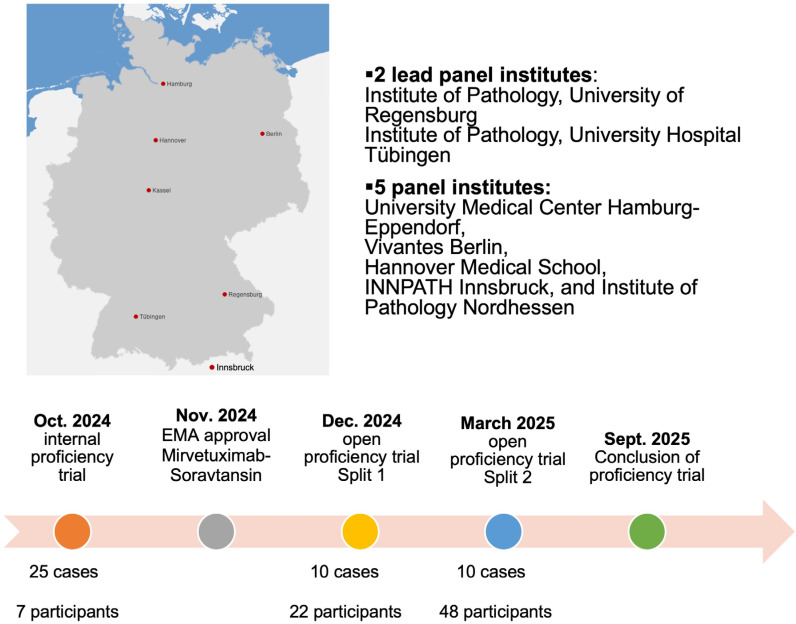
Flowchart depicting the time course of the internal and open proficiency trial.

**Figure 2 cancers-17-03703-f002:**
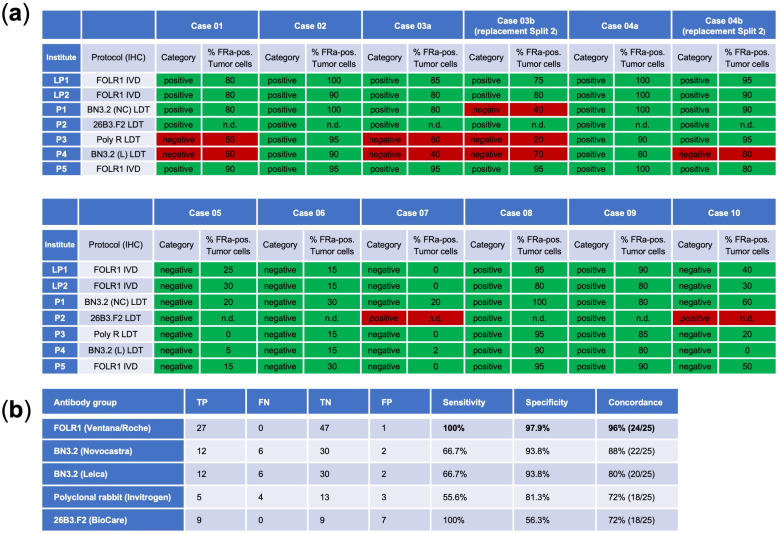
(**a**) Results of the Internal proficiency test for cases that were later also used in the open proficiency trial. Cases 03b and 04b represent replacement cases distributed in Split 2 (disregarded for scoring of the internal proficiency test). (**b**) Overview of antibodies utilized by the Lead and Panel Institute participants. TP, true positive; FN, false negative; TN, true negative; FP, false positive.

**Figure 3 cancers-17-03703-f003:**
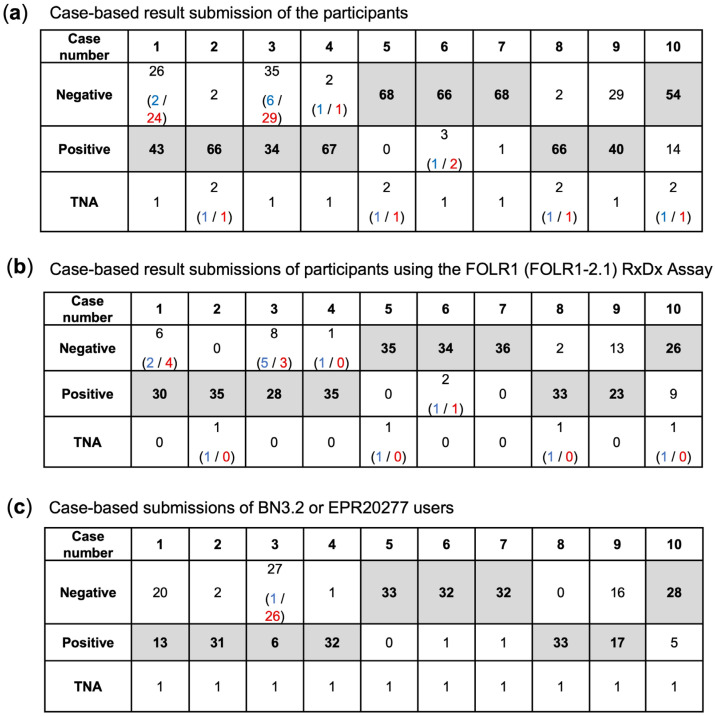
Case-based results for (**a**) all submissions of participants, (**b**) the submissions based on the FOLR1 RxDx Assay and (**c**) the submissions using BN3.2 or EPR20277 antibodies (the latter was utilized by only one participant). The tables show the number of participants who selected the corresponding answer. The target values are indicated in bold. The answers rated as correct are shaded in gray. In parentheses, the number of participants is given who provided responses deviating from the target value, and who, after the review process, were awarded points (blue text) or not awarded points (red text).

**Figure 4 cancers-17-03703-f004:**
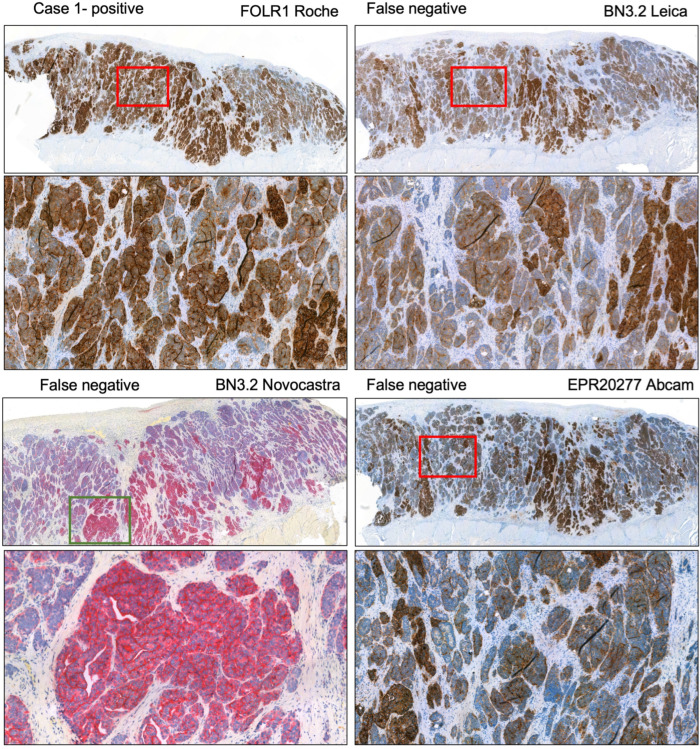
Case 1 assessed as positive by the lead panel institutes. Immunohistochemical stainings submitted by participants with different antibodies. The BN3.2 antibody clones as well as the EPR20277 Abcam antibody resulted in weak staining intensities that led to a false negative classification in the presented cases. Magnification 5x (overview) and 40x.

**Figure 5 cancers-17-03703-f005:**
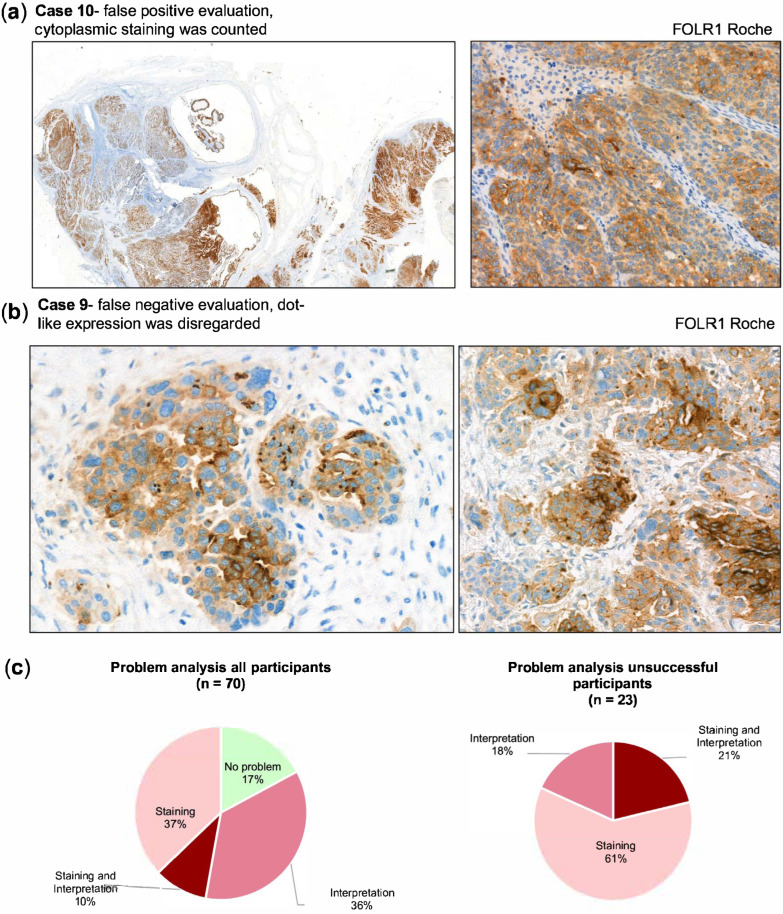
(**a**) Interpretation issue, where areas with cytoplasmic staining in tumor cells were wrongly counted as positive. Magnification left panel 5x, right panel 40x. (**b**) Example of an interpretation issue in case 9 where dot-like expression pattern (which is supposed to be counted as positive) was disregarded. Magnification both panels 40x (**c**) Problem analysis for all participants (left chart) and the participants that failed the proficiency test (right chart) with indication of underlying problem.

**Table 1 cancers-17-03703-t001:** Methods used by Lead panel and panel institutes in the internal proficiency test.

	Lead Panel Institute 1	Lead Panel Institute 2	Panel Institute 1	Panel Institute 2	Panel Institute 3	Panel Institute 4	Panel Institute 5
**Pretreatment**	CC1 (Ventana/Roche, Tucson, AZ, USA)	CC1 (Ventana/Roche, Tucson, AZ, USA)	EnVision™ FLEX Target Retrieval Solution, High pH (Dako/Agilent, Glostrup, Denmark)	EnVision™ FLEX Target Retrieval Solution, Low pH (Dako/Agilent, Glostrup, Denmark)	EnVision™ FLEX Target Retrieval Solution, High pH (Dako/Agilent, Glostrup, Denmark)	Citrate buffer	CC1 (Ventana/Roche, Tucson, AZ, USA)
**Antibody**	FOLR1 (FOLR1-2.1) RxDx Assay (Ventana/Roche, Tucson, AZ, USA)	FOLR1 (FOLR1-2.1) RxDx Assay (Ventana/Roche, Tucson, AZ, USA)	BN3.2 (Novocastra, Newcastle upon Tyne, UK)	26B3.F2 (BioCare Medical, Pacheco, CA, USA)	Polyclonal rabbit (Invitrogen, Carlsbad, CA, USA)	BN3.2 (Leica Biosystems, Newcastle upon Tyne, UK)	FOLR1 (FOLR1-2.1) RxDx Assay (Ventana/Roche, Tucson, AZ, USA)
**IVD assay/LDT**	CE IVD Assay FOLR1 (FOLR1-2.1) RxDx Assay (Ventana/Roche, Tucson, AZ, USA)	CE IVD Assay FOLR1 (FOLR1-2.1) RxDx Assay (Ventana/Roche, Tucson, AZ, USA)	Antibody: CE IVD marked—used as LDT	Antibody: CE IVD marked—used as LDT	Antibody: not CE IVD marked—used as LDT	Antibody: CE IVD marked—used as LDT	CE IVD Assay FOLR1 (FOLR1-2.1) RxDx Assay (Ventana/Roche, Tucson, AZ, USA)
**Dilution (primary Ab)**	RTU	RTU	1:150	RTU	1:500	1:100	RTU
**Incubation time (primary Ab)**	32 min	32 min	30 min	20 min	30 min	32 min	32 min
**Platform**	VENTANA BenchMark ULTRA (Ventana/Roche, Tucson, AZ, USA)	VENTANA BenchMark ULTRA (Ventana/Roche, Tucson, AZ, USA)	Dako Omnis (Dako/Agilent, Glostrup, Denmark)	Dako Omnis (Dako/Agilent, Glostrup, Denmark)	Dako Omnis (Dako/Agilent, Glostrup, Denmark)	VENTANA BenchMark ULTRA (Ventana/Roche, Tucson, AZ, USA)	VENTANA BenchMark ULTRA (Ventana/Roche. Tucson, AZ, USA)
**Detection system**	OptiView DAB IHC Detection Kit (Ventana/Roche, Tucson, AZ, USA)	OptiView DAB IHC Detection Kit (Ventana/Roche, Tucson, AZ, USA)	EnVision FLEX HRP DAB (Dako/Agilent, Glostrup, Denmark)	EnVision FLEX HRP DAB (Dako/Agilent, Glostrup, Denmark)	EnVision FLEX HRP DAB (Dako/Agilent, Glostrup, Denmark)	ultraView Universal DAB Detection Kit (Ventana/Roche, Tucson, AZ, USA)	OptiView DAB IHC Detection Kit (Ventana/Roche, Tucson, AZ, USA)

**Table 2 cancers-17-03703-t002:** Overview of success rates of internal and open proficiency tests.

	*n* Total	*n* Successful	Success Rate
**Internal proficiency test**			
All submissions	7	4	57%
**Open proficiency test**			
Split 1	22	12	55%
Split 2	48	25	52%
All submissions (open proficiency test, including repeated participations)	70	37	53%
Unique submissions (per institution)	63	37	59%
**Internal and open** **proficiency test combined**			
All submissions	77	41	53%
Repeated submissions (internal and open proficiency test)	9	6	67%
Unique submissions (per institution)	**68**	**41**	**60%**

**Table 3 cancers-17-03703-t003:** Overview of antibodies used by participants and case-by-case problem analysis.

Supplier	Antibody	*n* Participants	Successful Submissions	Case-by-Case Problem Analysis		
				Staining	Interpretation	Both
Roche Diagnostics	FOLR1 (FOLR1-2.1) RxDx Assay	36	30 (83%)	1	22	1
Leica Biosystems	BN3.2	23	5 (22%)	18	1	4
Novocastra	BN3.2	8	2 (25%)	4	2	2
Cell Signaling	BN3.2	1	0 (0%)	1	0	0
Abcam	EPR20277	1	0 (0%)	1	0	0
Invitrogen	Polyclonal rabbit	1	0 (0%)	1	0	0

## Data Availability

The data presented in this study are available on request from the corresponding author due to limitations regarding the anonymization of participating institutes.

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
