# Peer review of "Results of the First Folate Receptor Alpha Testing Trial by the German Quality Assurance Initiative in Pathology (QuIP®)"

_cancers, 2025, doi:10.3390/cancers17223703_

Round 1
Reviewer 1 Report
Comments and Suggestions for Authors
Scheiter and al. in this manuscript aimed to assess the validity to assess by IHC Folate receptor alpha (FRa) membrane staining comparing the VENTANA FOLR1 (FOLR1-2.1) RxDx 34 Assay, an assay already approved by FDA, respect to other FRa antibodies. The selection of standardized test among different institution to assess membrane FRa is a very important for the identification of ovarian cancer patients suitable for the therapy with Mirvetuximab soravtansine-gynx, an FRα-targeting ADC (antibody drug conjugated), when platinum-resistant, patients which have very poor therapeutic solutions. In addition, the use of an alternative cost effective assay to the expensive VENTANA FOLR1 (FOLR1-2.1) RxDx 34 Assay is desirable.
The proficiency trial presented in this manuscript is very important behind the final results not really succefull in the sense that only the VENTANA FOLR1 (FOLR1-2.1) RxDx 34 Assay was confirmed the best option. It is the first report taking together the efficiency of different antibodies to detect FRa membrane staining among a large number of different institutions. Although the methodologies with these alternative antibodies were not standardized among the different institutions in term of antigen retrival, antibody diluition, incubation time and detection systems, as also highlited by the Authors in the Discussion section, the data presented will allow the development of further studies applying more focused methodologies.
Some concerns are mainly related to the presentation of the data.
- Together with representative pictures of the 10 or 12 FFPE biopsies identified as positive cases with the VENTANA FOLR1 (FOLR1-2.1) RxDx 34 Assay, tubal epithelium and/or epithelium of distal tubules, known to highly express membrane FRa are worthy to be shown to give examples of ideal antibody reactivity. In the manuscript cases 1, 9 and 10 are shown but it is expected to see the best stainings.
- Are you sure that the punctate FRa staining is not representative of real FRa expression? FRa has been described to be also expressed in membrane rafts and/or associated to signalling molecules and therefore can be also find in particular membrane clusters.
- A separate paragraph with statistcs for evaluation of inter-observer variability need to be added.
- In the discussion section a comment about the incubation time could be also made especially for those antibodies with light staining but described by the companies as good antibodies.
Author Response
Scheiter and al. in this manuscript aimed to assess the validity to assess by IHC Folate receptor alpha (FRa) membrane staining comparing the VENTANA FOLR1 (FOLR1-2.1) RxDx 34 Assay, an assay already approved by FDA, respect to other FRa antibodies. The selection of standardized test among different institution to assess membrane FRa is a very important for the identification of ovarian cancer patients suitable for the therapy with Mirvetuximab soravtansine-gynx, an FRα-targeting ADC (antibody drug conjugated), when platinum-resistant, patients which have very poor therapeutic solutions. In addition, the use of an alternative cost effective assay to the expensive VENTANA FOLR1 (FOLR1-2.1) RxDx 34 Assay is desirable.
The proficiency trial presented in this manuscript is very important behind the final results not really succefull in the sense that only the VENTANA FOLR1 (FOLR1-2.1) RxDx 34 Assay was confirmed the best option. It is the first report taking together the efficiency of different antibodies to detect FRa membrane staining among a large number of different institutions. Although the methodologies with these alternative antibodies were not standardized among the different institutions in term of antigen retrival, antibody diluition, incubation time and detection systems, as also highlited by the Authors in the Discussion section, the data presented will allow the development of further studies applying more focused methodologies.
- We thank the reviewer for this thoughtful assessment of our work. The primary intent behind this study was to explore the freedom of methodologies in the context of folate receptor alpha (FRα) immunohistochemistry. Particularly outside of the United States, it is a general aim to enable testing approaches that, while not officially designated as companion diagnostics, should ideally provide results equivalent to those obtained with the companion diagnostic assays used in pivotal clinical trials. This flexibility is especially relevant within the European Union, where methodological freedom fosters competition among antibody manufacturers and encourages the development of cost-effective yet diagnostically reliable alternatives.
As the reviewer correctly pointed out, within the present folate receptor alpha proficiency trial, none of the alternative antibodies yielded results equivalent to the VENTANA FOLR1 (FOLR1-2.1) RxDx Assay by Roche Ventana. However, we would not characterize the trial as unsuccessful. On the contrary, it was successful in identifying a clear performance hierarchy and in demonstrating that, under current conditions, only the Roche Ventana assay provides reproducible and clinically reliable results across institutions.
We consider this outcome valuable, as it underscores the necessity and impact of such interlaboratory proficiency trials. These efforts are critical to objectively define analytical performance, establish current limitations, and lay the foundation for future rounds of proficiency testing that may eventually identify equivalent alternative antibodies or optimized staining protocols. While it remains desirable to find validated alternatives to the Roche assay, the present study provides important evidence that, as of now, no such equivalent option exists using the antibodies and detection systems tested.
Some concerns are mainly related to the presentation of the data.
Together with representative pictures of the 10 or 12 FFPE biopsies identified as positive cases with the VENTANA FOLR1 (FOLR1-2.1) RxDx 34 Assay, tubal epithelium and/or epithelium of distal tubules, known to highly express membrane FRa are worthy to be shown to give examples of ideal antibody reactivity. In the manuscript cases 1, 9 and 10 are shown but it is expected to see the best stainings.
- We thank the reviewer for this valuable suggestion. We fully agree that the inclusion of all 12 FFPE samples (including the two replacement cases for cases 3 and 4) will provide readers with a more comprehensive impression of the tested tissue and the heterogeneity observed among the cases. Accordingly, we have added an overview of all 12 cases as a Supplementary Figure 2, allowing readers to appreciate the full spectrum of staining intensity and heterogeneity as well as the interpretative challenges encountered.
In the main part of the manuscript, we continue to highlight cases 1, 9, and 10, as these are of particular relevance in illustrating the most frequent interpretative pitfalls. However, the supplementary figure now displays representative images from all included cases, selected from real-world submissions received from participating pathology departments in the proficiency trial. The examples shown represent correctly evaluated slides and illustrate staining patterns that led to successful performance in this trial. All images in the supplementary figure were generated using the standardized in-vitro diagnostic (IVD) Roche-Ventana assay (VENTANA FOLR1 [FOLR1-2.1] RxDx).
We also greatly appreciate the reviewer’s suggestion to include tubal epithelium as an example of strong physiological FRα membrane staining. In fact, several participating laboratories included sections of the tuba uterina epithelium on the same slides as internal quality controls. We have now added representative images of tubal epithelium to supplementary figure 2 and included a short paragraph in the manuscript (introduction) referring to their role as positive controls. These examples further demonstrate ideal staining performance and were helpful in achieving correct interpretation among participating institutions.
Are you sure that the punctate FRa staining is not representative of real FRa expression? FRa has been described to be also expressed in membrane rafts and/or associated to signalling molecules and therefore can be also find in particular membrane clusters.
- We thank the reviewer for this comment. The reviewer is perfectly correct in noting that folate receptor alpha (FRα) can localize within membrane rafts and signaling-associated membrane clusters, and therefore punctate staining patterns may indeed represent genuine FRα expression.
One should differentiate between two distinct morphological staining patterns: (i) dot-like or apical canalicular staining that corresponds to true membrane-associated FRα expression—particularly along small luminal structures—and (ii) focal punctate cytoplasmic staining, which does not represent genuine membranous expression and should therefore be disregarded.
We believe the reviewer refers to Figure 4, panel B (case 9), where we explicitly stated that a false-negative evaluation had occurred because the dot-like luminal expression was overlooked. We agree that such dot-like/apical canalicular staining should indeed be interpreted as positive, as it reflects genuine apical membrane expression. Importantly, in these cases, all tumor cells surrounding the luminal structure should be scored as positive, even if only a single central dot-like staining signal is present, as this corresponds to the apical part of the membrane. The corresponding section in our manuscript is:
„A further interpretative challenge was the recognition of dot-like expression patterns in narrow lumina. All tumor cells surrounding such luminal dot-like positivity should have been classified as positive, but several laboratories underestimated the proportion of positive tumor cells, which led to false-negative categorization (Figure 4).“
We think this section should be unambiguous and thus we did not change it.
A separate paragraph with statistcs for evaluation of inter-observer variability need to be added.
- We thank the reviewer for this valuable suggestion. The following paragraph was added to the results section:
To assess interobserver reliability, all submissions from participating laboratories were evaluated across the ten cases. Technically not assessable (TNA) results and values given in parentheses were excluded from the analysis.
The pooled analysis of all submissions against the reference classification showed an observed agreement of 83.4 % and an expected agreement by chance of 49.7 %, corre-sponding to Cohen’s κ = 0.67 (substantial agreement). The analysis of all submissions using the VENTANA FOLR1 (FOLR1-2.1) RxDx Assay showed an observed agreement of 88.5 % and an expected agreement by chance of approximately 51 %, corresponding to Cohen’s κ = 0.77 (substantial agreement). In contrast, the analysis of all submissions using the BN3.2 or EPR20277 antibodies showed an observed agreement of 77.9 % and an expected agreement by chance of 48.4 %, corresponding to Cohen’s κ = 0.57 (moderate agreement).
Moreover, a short decription was added to the methods section.
In the discussion section a comment about the incubation time could be also made especially for those antibodies with light staining but described by the companies as good antibodies.
- The incubation time was determined individually by each participating institution. While some guidance is provided by the respective antibody manufacturers in their product data sheets, these instructions serve only as general recommendations, as most of the antibodies used (except for the VENTANA FOLR1 RxDx Assay) are not IVD- or companion diagnostic–approved reagents.
In practice, laboratories are required to perform gradient testing and protocol optimization according to their staining platform and detection system. This typically involves adjustment of antibody dilution and incubation time, together with the use of appropriate positive control tissues.
For example, in Regensburg (one of the lead panel institutions), we initially optimized the Leica BN3.2 antibody by varying dilution and selected a working dilution of 1:50. However, at this dilution the achieved staining intensity did not reach the desired sensitivity, and further reduction would have diminished the economic advantage compared with the Roche RxDx assay.
Altogether, while this variation in incubation times and dilutions across institutions represents a limitation of our study, it also reflects a strength, since it robustly demonstrates a lack of sensitivity of the alternative antibodies, despite a large variety of testing setups and optimization attempts in diverse laboratory environments.
It should be noted that one participant (apparently aware of the sensitivity issue) using Leica BN3.2 even tried using a 1:10 dilution with 1 hour incubation time and still failed the proficiency trial.
We adjusted the discussion according to the suggestion:
Given that most participants already used low antibody dilutions (1:50 or 1:25) and one participant apparently aware of the lower sensitivity even went down to 1:10 and incubated for one hour still being unsuccessful, further dilution or incubation time adjustments are unlikely to improve performance.
Reviewer 2 Report
Comments and Suggestions for Authors
Chemotherapy resistance in ovarian cancer results from multiple recurrences in patients with advanced disease. The emergence of ADC drugs for FRα has prolonged the progression free survival (PFS) and overall survival (OS) of platinum-resistant patients with high FRα expression. In the United States, the VENTANA FOLR1 (FOLR1-2.1) RxDx assay was used to select FRα high-expression patients with ≥ 75% moderate-to-strong membrane staining in tumor tissues identified by immunohistochemistry (IHC). In the European Union, the flexibility of antibodies selection and detection systems is still maintained, and there is no unified standard requirement. So, the German Quality Assurance Initiative in Pathology (QuIP®) conducted this interlaboratory proficiency test to determine a reliable and reproducible immunohistochemical biomarker test.
In this manuscript, the authors evaluated the efficacy of different antibodies for detecting FRα through internal and open proficiency tests. The VENTANA FOLR1 RxDx Assay was found to have the highest reliability, while other alternative antibodies had lower concordance. Concurrently, they examined the underlying causes of failure in some cases and conducted a comparative analysis of the impact of the standard seminar.
Hence, the results of this study, in conjunction with the findings of internal and open proficiency tests, provide valuable insights into this topic area. It is suggested to revise the manuscript to justify the study design better and discuss findings in the context of existing studies, as follows.
Major comments:
- In the methods section, I think some additional details are required to add. Table 1 provides an overview of the antibodies and staining procedures used by the lead panel and panel institutions. It is not clear to me how the incubation time is determined. Was it based on each institution's experience, the kit instructions, or the optimal incubation time derived from gradient testing? Specifically, for antibodies requiring dilution, does the incubation time need adjustment when using different dilution ratios? Could this affect the institutions' results? The same question applies to the supplementation of Tables 3 and 4.
- The number of cases included in the study is relatively limited, and the sample size may not be sufficient to draw broader conclusions. Given the potential influence of individual variations, the findings might not be fully representative of a wider population. Although the researchers made efforts to address possible bias—such as by selecting replacement samples with FRα expression patterns similar to the original—some degree of variability is still expected to remain.
- We recommend including a flowchart in the Methods section to visually summarize the steps for internal and external testing. This will significantly help readers understand the experimental process.
- In Results 3.1 (Page 5, Lines 167-173), the analysis of Figure 1 discusses the causes of false negatives at panel institutes 3 and 4, as well as the false positives at institute 2. However, the analysis does not address the false negative result for Case 03b at panel institute 1. Could you comment on whether this was a chance occurrence, related to technical issues, or associated with the replacement sample? This information should be supplemented in the text. If no specific cause was identified, noting this fact would also be valuable for the readers.
Minor comments:
- On Page 5, Line 183, the text currently references "Supplementary Figure 1", but it should refer to "Supplementary Table 1". Please correct this.
- We note an inconsistency between the description of Figure 4 in the text and the actual figure labels. The text mentions "cytoplasmic staining" and "dot-like expression," but the order does not appear to match panels (b) and (c) in the figure. To prevent potential misunderstanding, please ensure the text description aligns with the figure, or consider adding a figure number (e.g., "Figure 4b," "Figure 4c") following the text.
- Page 10, Figure 4: The phrase “interpretation issues” below the figure (a) seems to serve as a descriptor for figure (b) and (c). It might be clearer to remove this phrase. You have already explained this in the figure caption.
Author Response
Chemotherapy resistance in ovarian cancer results from multiple recurrences in patients with advanced disease. The emergence of ADC drugs for FRα has prolonged the progression free survival (PFS) and overall survival (OS) of platinum-resistant patients with high FRα expression. In the United States, the VENTANA FOLR1 (FOLR1-2.1) RxDx assay was used to select FRα high-expression patients with ≥ 75% moderate-to-strong membrane staining in tumor tissues identified by immunohistochemistry (IHC). In the European Union, the flexibility of antibodies selection and detection systems is still maintained, and there is no unified standard requirement. So, the German Quality Assurance Initiative in Pathology (QuIP®) conducted this interlaboratory proficiency test to determine a reliable and reproducible immunohistochemical biomarker test.
In this manuscript, the authors evaluated the efficacy of different antibodies for detecting FRα through internal and open proficiency tests. The VENTANA FOLR1 RxDx Assay was found to have the highest reliability, while other alternative antibodies had lower concordance. Concurrently, they examined the underlying causes of failure in some cases and conducted a comparative analysis of the impact of the standard seminar.
Hence, the results of this study, in conjunction with the findings of internal and open proficiency tests, provide valuable insights into this topic area. It is suggested to revise the manuscript to justify the study design better and discuss findings in the context of existing studies, as follows.
- We thank the reviewer for this thoughtful assessment of our work and for giving us the opportunity to revise the manuscript accordingly. We highly appreciate the constructive feedback and will carefully address all points raised.
Major comments:
- In the methods section, I think some additional details are required to add. Table 1 provides an overview of the antibodies and staining procedures used by the lead panel and panel institutions. It is not clear to me how the incubation time is determined. Was it based on each institution's experience, the kit instructions, or the optimal incubation time derived from gradient testing? Specifically, for antibodies requiring dilution, does the incubation time need adjustment when using different dilution ratios? Could this affect the institutions' results? The same question applies to the supplementation of Tables 3 and 4.
- We thank the reviewer for addressing this important point. Regarding the Roche VENTANA FOLR1 [FOLR1-2.1] RxDx assay the incubation time was taken from the manufacturer’s instructions to conform with the IVD/ companion diagnostics protocol. Dilutions were not applicable in this case, since the antibody is ready to use.
Concerning the remaining antibodies, the incubation time was determined individually by each participating institution. While general guidance is provided by the respective antibody manufacturers in their product data sheets, these instructions serve only as broad recommendations, as most of the antibodies used (except for the VENTANA FOLR1 RxDx Assay) are not IVD- or companion diagnostic–approved reagents.
In practice, laboratories performed gradient testing and protocol optimization according to their specific staining platforms and detection systems. This typically included adjusting antibody dilution and incubation time in combination with appropriate positive control tissues. For example, at the lead panel institution in Regensburg, the Leica BN3.2 antibody was optimized by varying dilution ratios, and a working dilution of 1:50 was selected. However, this dilution did not achieve sufficient staining intensity, and further reduction would have diminished the economic advantage compared with the Roche RxDx assay (although Regensburg did not use the Leica BN3.2 clone for the ring trial, nonetheless we assessed it internally).
Overall, while variation in incubation times and dilutions across institutions represents a limitation, it simultaneously demonstrates robustness, as the lack of sensitivity of the alternative antibodies was consistently observed despite diverse optimization approaches across laboratories. Notably, one participant, apparently aware of the sensitivity issue, even used a 1:10 dilution with 1-hour incubation time and still failed the proficiency test using the Leica BN3.2 clone.
We have accordingly clarified this methodological aspect in the Methods section:
“For the Roche VENTANA FOLR1 [FOLR1-2.1] RxDx Assay, incubation time strictly followed the manufacturer’s instructions to ensure compliance with the companion di-agnostics protocol. As this antibody is supplied ready-to-use, no dilution adjustments were required. For all other antibodies, incubation times were individually determined by each participating institution. While general recommendations are provided in the re-spective antibody data sheets, these reagents are not IVD-approved, and therefore la-boratories performed gradient testing and protocol optimization according to their staining platforms and detection systems. This optimization typically involved fi-ne-tuning antibody dilution, incubation time, and detection chemistry using appropriate positive control tissues. Consequently, some variability in incubation times and dilutions existed across site reflecting real-world conditions.”
Moreover, we slightly adjusted the Discussion:
“Given that most participants already used low antibody dilutions (1:50 or 1:25) and one participant, apparently aware of the lower sensitivity, even reduced the dilution to 1:10 and incubated for one hour but still did not achieve sufficient staining, further adjustments in dilution or incubation time are unlikely to improve performance.”
- The number of cases included in the study is relatively limited, and the sample size may not be sufficient to draw broader conclusions. Given the potential influence of individual variations, the findings might not be fully representative of a wider population. Although the researchers made efforts to address possible bias—such as by selecting replacement samples with FRα expression patterns similar to the original—some degree of variability is still expected to remain.
- We thank the reviewer for raising this important concern and for giving us the opportunity to elaborate. Indeed, the total number of ten cases included in each split of the open proficiency test is limited. However, case selection followed predefined criteria ensuring a balanced representation of FRα expression levels, including clearly positive, clearly negative, and borderline (around the 75% cut-off) samples. This design reflects the real-world diagnostic spectrum and the challenges typically encountered in routine pathology, which is the key objective of an interlaboratory proficiency trial aimed at assessing both analytical robustness and interpretative concordance.
We acknowledge that the introduction of replacement samples could add minor variability; however, these cases were carefully selected from the same pool of 25 validated specimens to closely match the FRα expression patterns of the original cases. For transparency, representative images of all ten cases stained with the Roche VENTANA FOLR1 RxDx assay have been added to the supplementary material.
To further substantiate our findings beyond the limited sample number of the proficiency test, based on the reviewer’s comment we additionally validated the antibody performance using a self-made ovarian cancer tissue microarray containing both high-grade and low-grade serous carcinomas from primary and metastatic sites. This independent analysis, performed with the Roche VENTANA FOLR1 RxDx assay and the Leica BN3.2 antibody clone in two dilutions (1:200 and 1:50), confirmed the reduced sensitivity of the Leica clone compared to the IVD-certified Roche assay. Exemplary pictures from that microarray, now presented in Figure S2, support the reproducibility and generalizability of our main finding of reduced sensitivity of the BN3.2 clone from Leica. Moreover, it can be appreciated that an increase in antibody concentration does not remedy this issue, instead the cytoplasmic background gets more intense leading to a more difficult assessment of the membranous staining.
- We recommend including a flowchart in the Methods section to visually summarize the steps for internal and external testing. This will significantly help readers understand the experimental process.
- We agree with the reviewer, that a flowchart can help the reader understand the scope of our ring trial better. We added a respective flowchart to the Methods section.
- In Results 3.1 (Page 5, Lines 167-173), the analysis of Figure 1 discusses the causes of false negatives at panel institutes 3 and 4, as well as the false positives at institute 2. However, the analysis does not address the false negative result for Case 03b at panel institute 1. Could you comment on whether this was a chance occurrence, related to technical issues, or associated with the replacement sample? This information should be supplemented in the text. If no specific cause was identified, noting this fact would also be valuable for the readers.
- We thank the reviewer for this valuable observation. The false-positive result obtained at panel institute 2 using 26B3.F2 (BioCare Medical) is indeed noteworthy, as the majority of discordant results with the alternative antibodies tended to be false negatives rather than false positives. This finding can be attributed to a pronounced non-specific background staining, which obscured the true membrane signal and complicated accurate interpretation. Unfortunately, the corresponding slides are no longer available, and digital scans were not obtained at the time, so we are unable to include representative images as supplementary material. Accordingly, we have added the following clarification to the Results section:
“The false-positive interpretation was caused by pronounced non-specific background staining, which obscured the distinction between true membrane staining and background signal, thereby complicating accurate evaluation.”
Minor comments:
- On Page 5, Line 183, the text currently references "Supplementary Figure 1", but it should refer to "Supplementary Table 1". Please correct this.
- Thank you for pointing this out. We corrected this mistake.
- We note an inconsistency between the description of Figure 4 in the text and the actual figure labels. The text mentions "cytoplasmic staining" and "dot-like expression," but the order does not appear to match panels (b) and (c) in the figure. To prevent potential misunderstanding, please ensure the text description aligns with the figure, or consider adding a figure number (e.g., "Figure 4b," "Figure 4c") following the text.
- We chaged the order of the figure panels and added the exact subpanel number in the text.
- Page 10, Figure 4: The phrase “interpretation issues” below the figure (a) seems to serve as a descriptor for figure (b) and (c). It might be clearer to remove this phrase. You have already explained this in the figure caption.
- Thank you. We removed the phrase.
Round 2
Reviewer 1 Report
Comments and Suggestions for Authors
The Authors basically answered to my questions.